# A Metabolic Mechanism for Anaesthetic Suppression of Cortical Synaptic Function in Mouse Brain Slices—A Pilot Investigation

**DOI:** 10.3390/ijms21134703

**Published:** 2020-07-01

**Authors:** Logan J. Voss, Jamie W. Sleigh

**Affiliations:** 1Anaesthesia Department, Waikato District Health Board, Hamilton 3204, New Zealand; 2Department of Anaesthesia, Waikato Clinical Campus, University of Auckland, Hamilton 3204, New Zealand; jamie.sleigh@waikatodhb.health.nz

**Keywords:** anaesthesia, metabolism, mitochondria, synapse, mouse, cortex

## Abstract

Regulation of synaptically located ionotropic receptors is thought to be the main mechanism by which anaesthetics cause unconsciousness. An alternative explanation, which has received much less attention, is that of primary anaesthetic disruption of brain metabolism via suppression of mitochondrial proteins. In this pilot study in mouse cortical slices, we investigated the effect of disrupting cellular metabolism on tissue oxygen handling and cortical population seizure-like event (SLE) activity, using the mitochondrial complex I inhibitor rotenone, and compared this to the effects of the general anaesthetics sevoflurane, propofol and ketamine. Rotenone caused an increase in tissue oxygen (98 mmHg to 157 mmHg (*p* < 0.01)) before any measurable change in SLE activity. Thereafter, tissue oxygen continued to increase and was accompanied by a significant and prolonged reduction in SLE root mean square (RMS) activity (baseline RMS of 1.7 to 0.7 µV, *p* < 0.001) and SLE frequency (baseline 4.2 to 0.4 events/min, *p* = 0.001). This temporal sequence of effects was replicated by all three anaesthetic drugs. In conclusion, anaesthetics with differing synaptic receptor mechanisms all effect changes in tissue oxygen handling and cortical network activity, consistent with a common inhibitory effect on mitochondrial function. The temporal sequence suggests that the observed synaptic depression—as seen in anaesthesia—may be secondary to a reduction in cellular metabolic capacity.

## 1. Introduction

The idea that disruption to brain metabolism may be at least part of the causal mechanism of general anaesthesia is not new [1,2], but in recent times the hypothesis has been largely disregarded as research has focused on ion channel targets at synapses. Central to the metabolic hypothesis is the disproportionately large energy requirement for the brain to sustain consciousness [3]. Synaptic transmission, in particular, demands most of this energy [4]―and is consequently highly sensitive to metabolic stress [5,6]―providing an alternative pathway by which anaesthetics could act on the brain to disrupt consciousness.

Support for this hypothesis is more than just conjectural. Many volatile and intravenous anaesthetics have been shown to depress mitochondrial function [7,8], disrupting the mitochondrial membrane potential and reducing intracellular ATP levels [9]. In fact, anaesthetics have been proposed as pharmacological probes for investigating mechanisms of mitochondrial uncoupling [10]. In detailed work carried out over some decades, a strong case has been built for the specific respiratory chain protein complex I protein being the prime mitochondrial target. A striking sensitivity to volatile anaesthetics has been observed in children with mitochondrial deficits linked specifically to complex I [11]. Mice lacking a gene for a subunit of the mitochondrial complex I protein are similarly sensitive to volatile anaesthetics and to propofol [12]. In the same knockout model, isoflurane specifically inhibits mitochondrial complex I at lower concentrations than in wild types [13]. Thus, the possibility of a causal metabolic basis to general anaesthesia remains an open research question [9,14]. However, the question arises: “Exactly how does the ‘metabolic component’ of anaesthetic mechanism of action tie in with the competing body of work that concentrates on disturbance of synaptic function as the primary mechanism of anaesthesia?” To answer this, both (synaptic and metabolic) components need to be measured and manipulated in the same experimental preparation. 

We have developed a cerebrocortical model of general anaesthesia exploiting spontaneous, cortically generated paroxysmal activity in mouse brain slices exposed to zero-magnesium (no-Mg) artificial cerebrospinal fluid (aCSF). These so-called seizure-like events (SLEs) are driven primarily by facilitated glutamatergic synaptic transmission [15] and are suppressed by anaesthetics of all classes in proportion to their hypnotic potency [16]. This makes it an ideal model for interrogating how anaesthetics disrupt excitatory synaptic function to cause network-level changes in cortical population activity. 

The aims of this study were twofold. Firstly, to characterize the relationship between mitochondrial function, tissue oxygen and cortical population activity, using the mitochondrial complex I inhibitor rotenone. Thus, with rotenone, the temporal sequence that we hypothesized was, initially, an increase in tissue oxygen as the metabolic handling of oxygen was impaired [17], then, after some delay, as energy stores were depleted, an impaired ability of cortical networks to sustain population activity. The second aim was to compare these effects with a variety of different classes of general anaesthetics (sevoflurane, ketamine and propofol) to determine whether network-level anaesthetic effects in this model could be accounted for by a metabolic pathway. If the anaesthetic drugs were acting primarily to disturb synaptic function, we would expect any change in metabolic activity to occur coincidentally with (or after) the observed changes in synaptic activity. If the anaesthetic drugs were acting primarily on the metabolism, we would expect to see a decrease in synaptic activity occurring after the increase in extracellular oxygen.

## 2. Results

### 2.1. Single-Drug Effects

The effect of rotenone, a mitochondrial complex I inhibitor, on SLE activity and tissue oxygen, is illustrated in Figure 1 and Figure 2. The earliest effect was an increase in tissue oxygen, from a baseline of 98 (58) to 157 (87) mmHg during the period of rotenone exposure (*p* < 0.01). Importantly, the change in oxygen preceded any observed effect on SLE activity by about 10–20 min. During drug wash-out, tissue oxygen continued to increase further and was accompanied by a significant reduction in SLE root mean square (RMS) activity (baseline RMS of 1.7 to 0.7 µV, *p* < 0.001) and SLE frequency (baseline 4.2 to 0.4 events/min, *p* = 0.001). In one slice tested, neither population activity nor tissue oxygen levels recovered after an extended wash period of 2.5 h. These delayed and prolonged changes indicate that the rotenone effect is slow and long lasting (perhaps irreversible). Rotenone also caused a qualitative change in SLE activation from a regular tonic pattern to a burst pattern characterized by long silent periods interspersed by bursts of varying length. This pattern change was evident in 9/10 cases. 

The anaesthetic effects are summarised in Figure 2. Similar to rotenone, the earliest effect of ketamine and sevoflurane was an increase in tissue oxygen, prior to any change in SLE RMS activity or event frequency. The trend was similar for propofol, although the oxygen increase compared to baseline was only significant during the drug wash-out period. All three anaesthetics strongly inhibited SLE frequency during the drug wash-out period, with a reduction of 25% for sevoflurane (*p* = 0.001), 76% for ketamine (*p* = 0.001) and 75% for propofol (*p* < 0.001). RMS activity during the drug wash-out period was variably affected (see Figure 2). These delayed effects on event frequency and RMS reflect a prolonged action that outlasts the presence of the drug in the aCSF—and probably the tissue. Estimates of the tissue drug level time courses are presented in Figure 3. They show that, with the exception of propofol, tissue drug levels return to near baseline levels within 15 min of beginning the drug wash-out. The prolonged presence of propofol is consistent with its slow diffusion in avascularised brain tissue [18] and is reflected in the ongoing bursting activity evident in Figure 4. The bursting pattern of activity was most strongly evident with propofol (11/12 cases) and sevoflurane (8/11 cases), but less so for ketamine (4/10 cases). The slow diffusion of propofol into the tissue probably also explains the slower initial increase in oxygen compared to the other agents.

As a control, switching perfusion from no-Mg aCSF (high metabolic demand) to normal aCSF (lower metabolic demand) resulted in a reversible increase in tissue oxygen, accompanied simultaneously by a strong reduction in RMS activity (see Figure 5). This differs from the drug effects described above, which showed a clear temporal disconnection between the tissue oxygen effects and changes in SLE RMS. The increase in tissue oxygen following the switch from no-Mg to normal aCSF was most likely a direct consequence of the reduced tissue metabolic demand.

### 2.2. Combined Drug Effects

Each of the anaesthetics was tested in combination with rotenone to determine if combined exposure had an additive effect. Propofol and sevoflurane showed heightened combined responses. Specifically, the propofol–rotenone combination caused a biphasic frequency response, with an initial increase during drug delivery (36 (44)% increase, *p* < 0.05), followed by an enhanced decrease during the drug wash period (a 75 (28)% decrease with propofol compared with a 95 (6)% decrease combined, *p* < 0.001). The sevoflurane–rotenone combination induced a significant reduction in event frequency during the drug delivery period (38 (37)% decrease, *p* < 0.01) and enhanced the decrease in RMS during the drug wash (an 8 (38)% decrease with sevoflurane compared with a 76 (22)% decrease in combination, *p* < 0.001). In each case highlighted above, there was no statistical difference between individual drug effects. The change in tissue oxygen was not enhanced in any combination. Nor was the ketamine effect enhanced according to any of the measures.

## 3. Discussion

In this pilot study, we have investigated the effect of disrupting cellular metabolism on cortical population activity and tissue oxygen, using the mitochondrial complex I inhibitor rotenone, and compared this with the effect of the general anaesthetics sevoflurane, propofol and ketamine. The rotenone results reflect the expected temporal sequence of tissue effects for a mitochondrial inhibitor. Firstly, a disruption to mitochondrial function results in an increase in tissue oxygen as the metabolic handling of oxygen is impaired [17]. Subsequently, when energy stores are depleted, there is an impaired ability of cortical networks to sustain population activity, resulting in the observed reduction in SLE RMS activity and event frequency. Suppression of cortical function outlasted the presence of the drug in the aCSF by at least 2 h, suggesting either a prolonged effect or irreversible damage perhaps due to reactive oxygen species generation [17]. 

Broadly speaking, the tested anaesthetic drugs produced effects whose time course mirrored those of rotenone, most notably an increase in tissue oxygen prior to a decrease in SLE activity, but also, after a lag of about 10–20 min (depending on the agent), a strong reduction in event frequency. Contrast these findings with that of magnesium, which inhibits excitatory synaptic function by directly blocking NMDA receptors [15]. Wash-in of magnesium caused temporally synchronous changes in tissue oxygen and population activity, indicating that the increase in tissue oxygen in this case was driven by a physiological reduction in metabolic demand, secondary to a primary reduction in synaptic activity. If the anaesthetics tested in this study were acting directly and solely at the level of synaptic receptors, a similar synchronization between tissue oxygen and population activity would be expected. Drawing a parallel mechanistic explanation with that of rotenone, we suggest that the observed anaesthetic synaptic depression is causally linked to a disruption in the mitochondrial handling of oxygen. These findings in no way discount direct synaptic receptor effects, which are clearly important. For example, point mutations to the GABA_A_ receptor markedly reduce the potency of propofol and etomidate in mice [20,21]. Importantly, these GABA_A_ mutations did not completely abolish anaesthesia, implicating involvement of additional mechanisms in the hypnotic effect. We therefore suggest a parallel indirect mechanism for anaesthetic suppression of synaptic function. The existence of this additional mechanism could also explain the lack of efficacy of direct synaptic antagonists to reverse general anaesthesia with propofol, or sevoflurane, or ketamine. 

It is important to note that we have not directly measured mitochondrial function in this study and that the specific target of the metabolic anaesthetic effects described herein is not identified by this study. While we note that rotenone is a specific complex I inhibitor—and that the rotenone effect was enhanced by sevoflurane and propofol—this does not necessitate complex I as the anaesthetic target. More specific experiments linking mitochondrial function are required to identify the effect mechanism, for example, using the Ndufs4 complex I knockout model [12,13].

An important question is that of effect delay due to drug diffusion into the active region of the slice, and whether this could account for the timing disconnect between the changes in oxygen levels and electrophysiological activity. If we take an opposing position, and assume that the increase in tissue oxygen is due to a drug-induced disruption to slice metabolism—but that this disruption is not mechanistically linked to the change in SLE activity—then a process facilitating faster drug action at intracellular mitochondria than at synaptic receptors would be necessary to account for the more rapid change in oxygen levels. Given the diffusion characteristics of these drugs and their rapid action at ionotropic synaptic receptors [22], this is difficult to envisage. We would also note that anaesthetic ethers (such as sevoflurane) and hydrocarbons diffuse at a comparable speed to magnesium in avascular brain slice tissue (see Figure 3). More likely, the change in tissue oxygen levels (reflecting mitochondrial impairment) is causally linked to the subsequent change in SLE activity. 

Supporting this is the finding that propofol and sevoflurane effected stronger changes in SLE activity when combined with rotenone, over and above the effects of either drug on their own. On the other hand, ketamine combined with rotenone did not show an enhanced effect. Ketamine is an enigmatic drug with a mechanism of action that is difficult to reconcile with conventional paradigms used for understanding anaesthetic mechanisms. This may also be the case here. While preclinical rodent and human cell culture studies have suggested ketamine may impair mitochondrial function [23,24], the neurobehavioural outworking is quite different from propofol and volatile anaesthetics [12]. Whereas complex I inactivation increases sensitivity to propofol and isoflurane, ketamine potency is actually reduced in this model [12].

A common effect seen with rotenone and the anaesthetics (especially propofol and sevoflurane) was a qualitative change in the pattern of SLE activation, from a regular tonic pattern to a burst pattern characterized by long silent periods interspersed by activity trains of varying length. This is similar to what has been observed previously with propofol in hippocampal slices [25] and may reflect a tendency for these drugs to promote ictal activity [26]. These bursts were often quite intense (especially for propofol), which is why RMS activity did not always decrease during the bursting period (despite a strong reduction in burst frequency). This accounts for some of the variation in RMS activity between agents during the wash-out period.

One question about our results might be that, in vivo, these anaesthetic drugs are known to have a very rapid onset of action. Drug action in avascularised slice preparations such as ours is much slower because drug movement is dependent upon passive diffusion from the slice surface. This is seen in the peak average effect on neural activity, which occurred after beginning the drug wash-out period (see Figure 2 results). These experiments were also done at room temperature, and the cortex is not activated by brain stem neuromodulators, which greatly reduces tissue metabolic demands. In fact, an advantage of our slice model is that the avascularity and low metabolic demand in this preparation has the effect of stretching out and adding greater resolution to the time course of the anaesthetic effects. Whether the metabolic effects of anaesthetics are sufficiently rapid to contribute meaningfully to loss of consciousness in the clinical setting is not clear, although it can be noted that, once at the mitochondria, the effect of isoflurane is almost immediate [27]. It may be that delayed emergence is a more clinically important consequence of mitochondrial impairment, along with postoperative sequelae such as delirium and cognitive decline.

In conclusion, anaesthetics with differing synaptic receptor actions effect changes in tissue oxygen handling and cortical network activity consistent with a common inhibitory effect on mitochondrial complex I function. Synaptic depression secondary to an indirect metabolic pathway is a likely mechanism of anaesthetic action. 

## 4. Materials and Methods

The tissue recovery methods in this study were approved (#1062, February 2019) by the Animal Ethics Committee at the University of Waikato, New Zealand. 

### 4.1. Tissue Preparation

Adult male or female C57 mice were anaesthetised with CO_2_ and the brain rapidly dissected and submerged in ice-cold “normal” aCSF. The brain was sliced coronally in ice-cold normal aCSF into 400-µm-thick sections between Bregma 1 to -5 using a vibrotome (Campden Instruments Ltd., Sileby, Leics, UK). Subsequently, the slices were immersed in aCSF void of magnesium (no-Mg aCSF) at room temperature (approximately 21 °C) for at least 60 min. Exposure to no-Mg aCSF activates the tissue by unblocking NMDA receptors [28], resulting in the generation of repeating, spontaneous paroxysmal events known as seizure-like events (SLEs). 

### 4.2. aCSF Solutions

Normal aCSF was composed of 125 mM NaCl, 2.5 mM KCl, 1mM MgCl_2_, 2 mM CaCl_2_, 1.25 mM NaH_2_PO_4_, 2.5 mM NaHCO_3_, 10 mM HEPES and 10mM D-glucose. No-Mg aCSF was composed of 124 mM NaCl, 5 mM KCl, 2 mM CaCl_2_, 1.25 mM NaH_2_PO_4_, 2.5 mM NaHCO_3_, 10mM HEPES and 10 mM D-glucose. All solutions were made in double-distilled water and oxygenated prior to use (95%, Perfecto2, Invacare, Auckland, New Zealand).

### 4.3. Data Recording

The slices were transferred one at a time to a submersion-style perfusion bath (Kerr Scientific Instruments, Dunedin, New Zealand), which was continuously flowed with no-Mg aCSF at a rate of 5 mL/min. A nylon net was loosely positioned above the slice to stabilize the tissue and help maintain a stable bath solution level. No-Mg SLE activity was recorded using four 75-µm Ag/AgCl electrodes inserted into the cortex, one within each quarter section (Figure 6). A disc Ag/AgCl electrode positioned distant to the slice in the recording bath was used as a common reference/ground. The electrodes were positioned extracellularly to monitor field potential. The analogue signal was amplified (Model 3000 differential amplifier, A-M Systems, Sequim, WA, USA) and converted to a digital signal (PowerLab, ADInstruments, Bella Vista, Australia) for later analysis. The analogue signal was filtered with: high pass—1 Hz, low pass—300 Hz and a notch filter at 50 Hz. The gain was ×1000 and the sampling rate 1000/s.

### 4.4. Oxygen Measurement

Tissue oxygen (PO_2_) was measured using a Clark-style oxygen electrode a with tip diameter of 25 µm (Unisense Ltd., Aarhus, Denmark). The electrode was polarized at the beginning of each day to achieve stable output. A two-point calibration was carried out in aCSF equilibrated with room air and in a solution of 0.1M sodium ascorbate (zero-point). Calibration was repeated at the beginning of each experimental day. The oxygen sensor was positioned in cortical layer III-IV to a depth of approximately 200 µm using a precision micromanipulator (FX-117, Minitool Inc., Campbell, NC, USA). No particular cortical region was targeted, and the final location was largely dependent upon where the nylon top-net allowed unimpeded access to the mid-cortical layers. The oxygen signal was recorded continuously at a 5-Hz sampling rate. 

### 4.5. Drug Preparation and Delivery

All drugs were added directly to no-Mg aCSF from stock solutions. Rotenone (Sigma, NZ) was delivered at 1.5 µM; propofol (Provive, Claris, Ahmeddabad, India) 56 µM; ketamine (Ketalar, Hospiro Melbourne, Australia, Pty Ltd.) 17 µM; sevoflurane (Sevorane, Abbott Laboratories, Maidenhead, UK) approximately 1MAC. Sevoflurane was delivered by adding 0.08 mL liquid to 200 mL no-Mg aCSF in a semi-sealed vessel. This delivery method was based on a previous investigation showing that 0.01 mL isoflurane added to 50 mL aCSF and perfused via a syringe pump at a rate of 5 mL/min generated a slice bath isoflurane concentration close to 1 rat MAC [29]. For this study, the sevoflurane amount was doubled to account for its MAC value being twice that of isoflurane. The effect on SLE activity was qualitatively identical to that of isoflurane (see Figure 7 and [30]), confirming approximate 1MAC equivalence. The concentration of rotenone (1.5 µM) was chosen following a pilot study showing that this dose effected an increase in tissue oxygen, along with quantifiable, but not complete, suppression of SLE activity. A concentration of 0.5–1.0 µM had no effect on SLE activity, while 3–10 µM rapidly and irreversibly suppressed all activity (results not shown). The propofol and ketamine concentrations were the same as those used previously in in-vitro cortical slice investigations [25,31].

### 4.6. Drug Concentration Time Course Estimation

The drug perfusion bath concentration time course was estimated by measuring the real-time (0.2-Hz sample rate) change in bath conductivity while perfusing a 0.1-mM solution of sodium chloride. The sodium chloride solution was perfused from a starting concentration of 0 mM, with the same flow parameters used for drug delivery. Because solution conductivity increases linearly with increasing salt concentration [32], the conductivity time course is a reasonable estimate of the drug concentration time-course. The tissue drug levels were estimated by applying an effect-site compartment and inhibitory E_max_ model describing the relationship between drug concentration and SLE frequency (see [19] for details). This method could not be used for rotenone because of its irreversible effect on event frequency over the recorded time frame.

The propofol concentration was based on a clinically relevant concentration at loss of consciousness of 2 µM [33] and the slow diffusion of propofol into isolated brain slices [18]. It can be estimated that the average propofol concentration between a depth of 25–200 µm in a 400-µm-thick slice is approximately 0.1 of the aCSF concentration after 15–20 min [18]. 

### 4.7. Experimental Protocols

Each experiment was preceded by a baseline recording period in which stable SLE activity was recording for at least 10 min in no-Mg aCSF. Thereafter, no-Mg aCSF containing rotenone (*n* = 9, from 4 animals), propofol (*n* = 6, from 4 animals), ketamine (*n* = 6, from 4 animals), or sevoflurane (*n* = 5, 1 animal) was perfused. With the exception of ketamine (10 min), the drugs were perfused for 15 min, followed by wash-out with drug-free no-Mg aCSF for 30 min. In addition, separate experiments were conducted combining rotenone with each of the anaesthetics (propofol *n* = 6, 2 animals; ketamine n = 6, 1 animal; sevoflurane *n* = 6, 2 animals).

Control experiments investigating the effect of changes in tissue metabolic rate on intra-slice oxygen levels and SLE activity were conducted by switching from no-Mg (high slice metabolic activity) to normal aCSF (low slice metabolic activity) (*n* = 4, 3 animals). 

### 4.8. Data Analysis and Statistics

Initially, the four channels recorded from each slice were visually inspected. Recording locations where SLE activity was clearly desynchronized between locations were treated as independent datapoints for statistical analysis. For locations where SLEs were synchronized (reflecting the spatial spread of a common wave front), data were averaged across locations and included as a single datapoint for statistical analysis. Further details and explanation of the basis of this delineation can be found in a previous investigation [34].

Changes in SLE activity were quantified in two ways: as the RMS voltage and as event frequency. The former effectively estimates the absolute value of the area under the waveform and was used here as an overall measure of neuronal activity. This provided a measure of the intensity of neuronal activity that was less dependent on event frequency alone. RMS was first calculated for the entire recording. A segment of inter-event background noise was then manually selected, and this RMS level subtracted from the recording as a whole. As such, only SLE activity above background noise was quantified. RMS values were then averaged across the “baseline” period (5 min prior to drug delivery), during drug delivery and during drug wash-out. Slice oxygen levels and event frequency were averaged over the equivalent time periods. The latter was defined as the inter-event frequency (distinct from the oscillatory frequency within a single event). In all cases, seizure-like bursts separated by suppression periods greater than approximately half the burst length were quantified as single events. Long bursting activity was evident for all tested drugs, but was particularly strong for propofol (see Figure 4 and [25]). 

Differences between three paired groups were assessed for statistical significance using the repeated measures ANOVA or Friedman test (for normally and non-normally distributed data, respectively). For unpaired data, one-way ANOVA and Kruskal–Wallis tests were used. Tukey and Dunn post-tests were utilized for normally and non-normally distributed comparisons, respectively. Differences between two paired groups were assessed for statistical significance using the paired t-test or Wilcoxon test (for normally and non-normally distributed data, respectively). A *p*-value < 0.05 was considered statistically significant. Data are presented as mean (SD) unless stated otherwise.

## Figures and Tables

**Figure 1 ijms-21-04703-f001:**
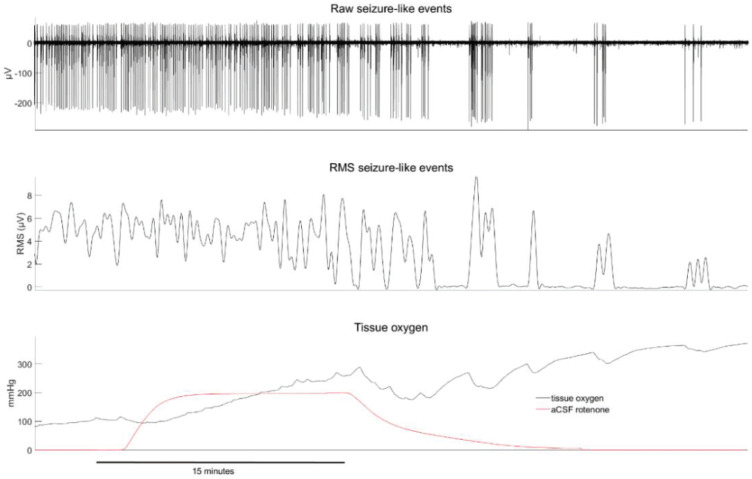
Example of rotenone-induced changes in seizure-like event activity (top), root mean square (RMS) seizure-like activity (middle) and tissue oxygen (bottom). The red line in the bottom figure shows the estimated concentration time course of rotenone in the perfusion bath (normalized to its maximum value and ×200 for visual scaling). Note that the tissue oxygen levels fluctuate with the intensity of neuronal activity. This is particularly evident in the second haft of the recording where bursting activity is punctuated by long silent periods.

**Figure 2 ijms-21-04703-f002:**
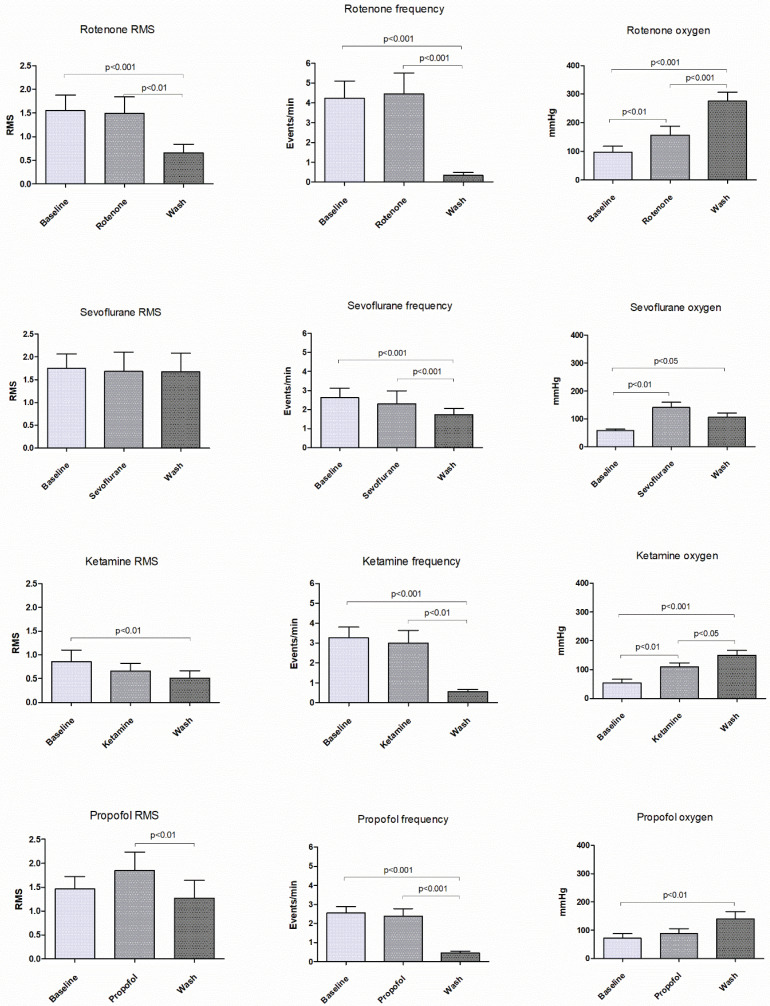
Graphs illustrating the effects on root-mean-square (RMS) seizure-like event (SLE) activity (left column), SLE frequency (middle column) and tissue oxygen (right column) of rotenone (1.5 µM), sevoflurane (1 MAC), ketamine (17 µM) and propofol (56 µM). Data are mean + SEM.

**Figure 3 ijms-21-04703-f003:**
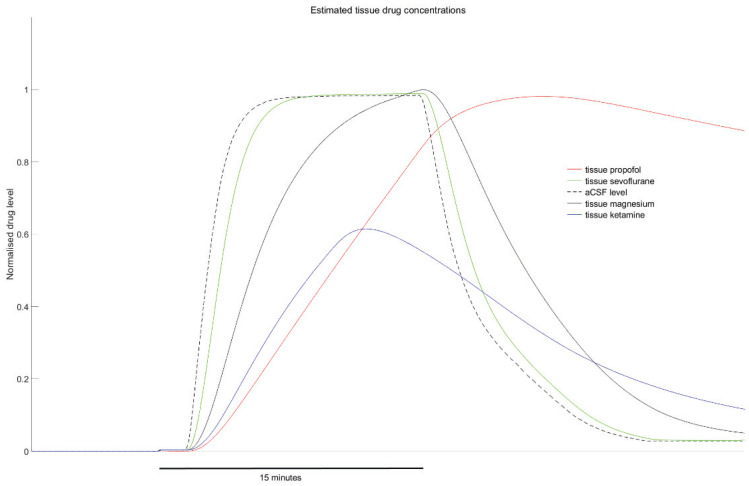
Normalized time course estimates for drug movement into and out of the tissue slices. The dotted black line indicates the aCSF drug level time course for a 15-min wash-in (note that the ketamine wash-in period was only 10 min). The tissue levels were estimated by applying an effect-site compartment and inhibitory E_max_ model describing the relationship between drug concentration and seizure-like event frequency (see [19] for details).

**Figure 4 ijms-21-04703-f004:**
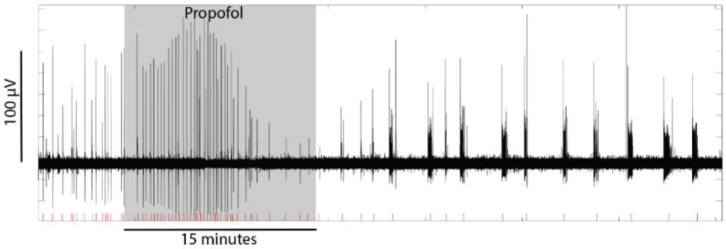
Example from one slice showing the effect of propofol on seizure-like event (SLE) activity. The shadowed region represents the period of drug delivery. The red lines in the figure illustrate how event frequency was quantified—in particular, that distinct bursts of activity were counted as single events.

**Figure 5 ijms-21-04703-f005:**
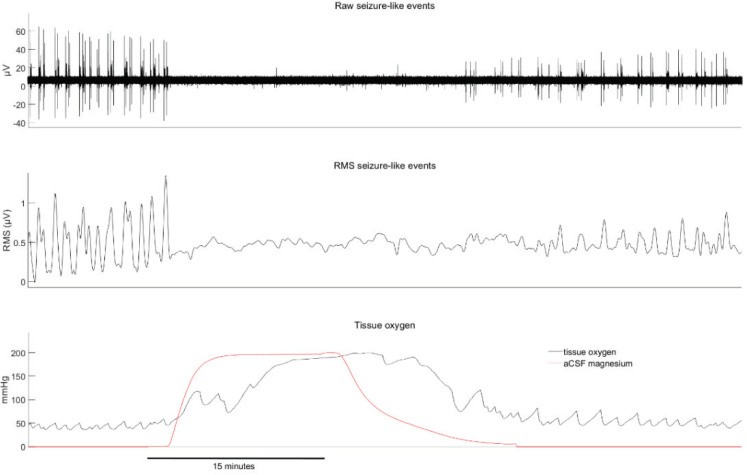
Example of changes in seizure-like event activity (top), root mean square (RMS) seizure-like activity (middle) and tissue oxygen (bottom) with the transition from no-Mg aCSF to normal aCSF and back to no-Mg (i.e., wash-in of magnesium ions). The red line in the bottom figure shows the estimated concentration time course for magnesium in the perfusion bath (normalized to its maximum value and ×200 for visual scaling).

**Figure 6 ijms-21-04703-f006:**
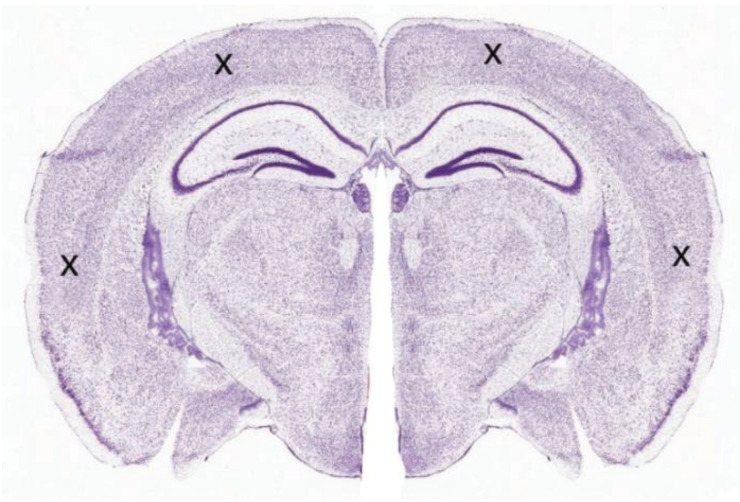
Coronal rodent slice showing the approximate recording electrode arrangement (marked by the crosses).

**Figure 7 ijms-21-04703-f007:**
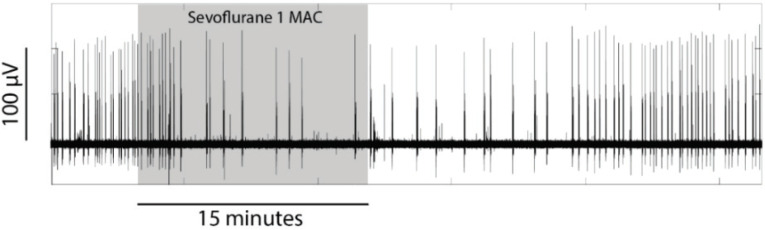
Example from one slice showing the reversible inhibitory effect of sevoflurane on seizure-like event (SLE) activity. The shadowed region represents the period of drug delivery.

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
