# Peer review of "A Metabolic Mechanism for Anaesthetic Suppression of Cortical Synaptic Function in Mouse Brain Slices—A Pilot Investigation"

_ijms, 2020, doi:10.3390/ijms21134703_

Round 1

Reviewer 1 Report

Manuscript: ijms-804391

Title: A primary metabolic mechanism for anaesthetic suppression of cortical synaptic function in mouse brain slices

Authors: Voss LJ, Sleigh JW

General.  This is an interesting article that postulates the root cause of general anesthesia (GA) is at least partially due to a metabolic effect.  In a seizure model using mouse brain slices (in low magnesium), they show that tissue oxygen tension rises (and therefore presumably consumption falls) prior to observable changes in neuronal function.  They show this fairly convincingly, although there are a couple concerns (listed below).  The model is rather indirect and they overstate the strength of their proof, in my opinion.  However, they do generate data that is consistent with the metabolic model.  Taken as a whole, the work seems a bit more like a strong pilot study needing some further work (see #6).

Specific concerns.

  1. While the authors do reference some data suggesting metabolic models for GA, they really miss the rather rich literature during the past 20 years suggesting a mitochondrial target for at least the volatile anesthetics.  The first reference (#7) is a clinical review and really not helpful here. They should examine the works of Sedensky, Zimin, Perouansky, Kayser in elegans, Drosophila, and mouse which strongly supports their model.  None of that is included.
  2. Their reference of the role of ketamine on mitochondrial function was found to be flawed and they should review the responding letter to those authors for clarification.
  3. It makes no sense to me that sevoflurane should only affect frequency during washout and not during exposure, This absolutely needs to be directly addressed.  In that vein, they have not taken residual glycolysis into consideration.  Could the delay in neuronal function simply be the time required to remove glucose/glycogen effects from glycolysis and leave the cells dependent on oxidative phosphorylation.  This could be tested with glycolysis inhibitors.
  4. The authors do not explain the importance of RMS (power?) compared to frequency. For many readers such a discussion would be very helpful.
  5. Since the authors have slices and are interpreting the oxygen levels to indicate metabolic changes, why do they not do a direct measurement of ATP levels. This would greatly strengthen the manuscript.
  6. The Ndufs4 KO mouse is available. The authors need to repeat this experiment with a genetic model such as Ndufs4.  If their model is correct, then the oxygen changes should occur at much lower sevoflurane concentrations in the mutant.  Of course, that would take significant time, but should at least be stated as in their plans to prove the model.  Until that is done, they cannot invoke a complex I mechanism, in my opinion. However, if that works, they would have a very strong model.
  7. Finally, I am surprised that the authors did not perform a dose response curve for sevoflurane to find a concentration where O2 rises, but activity remains high. If their model is correct, it should be possible to find a steady state concentration that separates the effects.  This would seem to be necessary before reaching their conclusion, at present one could still construct pharmacokinetic explanations for their findings.

Reviewer 2 Report

Review of the manuscript entitled „A primary metabolic mechanism for anaesthetic suppression of cortical synaptic function in mouse brain slices” by Voss & Sleigh for the International Journal of Molecular Sciences

In their well written manuscript Voss & Sleigh revitalize the idea of a metabolic mechanism of general anaesthetics by comparing the actions of the piscicide rotenone and several general anaesthetics on neuronal firing patterns in acute brain slices.

Major comments:

1.The changes in firing patterns of cortical neurons observed after the wash-in of anaesthetics like propofol or sevoflurane are absolutely convincing. A similar pattern of long bursting activity combined with a prolongation of silent periods was described in cortical neurons (although cultivated!) for propofol (Drexler et al., 2009) and sevoflurane (Grasshoff et al., 2007) before. However, the authors might want to comment on the concentrations of drugs used for their study as well as on the wash-in- and diffusion-times, respectively.

Concentrations: The concentrations of almost all the drugs used for the current study seem quite a bit high. For propofol ex vivo aqueous concentrations of 0.5 to 1 µM have been proposed, as these concentrations correspond to loss of consciousness and loss of righting reflex in vivo, respectively, (Franks, 2008). The current study uses 56µM. Rotenone has been used in acute brain slices at concentrations starting from 5 nM (Freestone et al., 2009). Unfortunately, the authors do not give an exact concentration of sevoflurane in mM in their manuscript.

Wash-in-times, diffusion-times: The authors state that their Clark-style oxygen electrode was positioned in a depth of roughly 200 µm of the 400 µm slice. Therefore, it seems reasonable to assume that also their Ag/AgCl electrodes were approximately that deep. In figure 3 the authors estimate the time course of drug movement into the slice. However, it remains unclear, what this estimate is based on. Benkwitz and co-workers stated that “diffusion of etomidate into brain slices requires approximately an hour to reach 80% equilibration at a typical recording depth of 100 µm” (Benkwitz et al., 2007). Judging from figure 5D of that paper this takes even longer for a recording depth of 200 µm. Admittedly, this study was on etomidate while the current study is on propofol. Still, Gredell and co-workers (Gredell et al., 2004) (reference # 18 in the present Voss & Sleigh manuscript) similarly report that “equilibration requires several hours” for propofol in brain slices.

Taken together, would it be possible, that the seemingly high concentrations used for the current study are in fact lower in a recording depth of approximately 200 µm and that the estimates presented in figure 3 might be too optimistic?

2. From a clinical point of view: When we inject e.g. propofol into a patients vein, loss of consciousness occurs quite fast, considering that the drug has to travel the vein, the right heart, the lung, the left heart and reach the brain. As we already know about synaptic actions of general anaesthetics (Alkire et al., 2008) it might be hard to believe that a primary metabolic mechanism significantly contributes to that particular action of general anaesthetics. But what about hysteresis observed in anaesthesia (Baars et al., 2006; Voss et al., 2012)? Might this be related to metabolic mechanisms? What about neurotoxic actions of general anaesthetics? What about delirium and POCD?

3. A primary metabolic mechanism for anaesthetics is indeed an intriguing idea. But how does that work? Is there any idea what the mechanism behind that might be? We know that, e.g. that benzodiazepines not only act (fast) via GABAA receptors, but also (perhaps slower) via TSPO, a large protein in the outer mitochondrial membrane (Rupprecht et al., 2010). Is anything known about similar actions of general anaesthetics? Perhaps the authors would like to comment/speculate in the discussion section of their manuscript.

4. I wonder why the authors address the firing patterns they observe in cortical slices as “seizure like”. From my point of view the concept of cortical up- and down- states has gained considerable importance in the past years, as indicated by several papers published on this issue (for example: (McCormick et al., 2003; Shu et al., 2003b; Shu et al., 2003a; Lau and Bi, 2005; Johnson and Buonomano, 2007). These papers refer to the phenomenon that activity in isolated cortical tissue is characterized by phases of activity – termed burst or up state – separated by phases of relative neuronal quiescence (termed down states). This is not an artefact, but characteristic of deafferented neocortex in general, as both acute slices (Sanchez-Vives et al., 2010) and deafferented neocortex in vivo (Timofeev et al., 2000) engage in activity patterns very similar to those seen in cortical slices and cultures.

Furthermore, the firing pattern of cortical slice cultures is typically characterized by a sequence of up states and down states (Crain and Bornstein, 1964; Plenz and Aertsen, 1996; Klostermann and Wahle, 1999; Johnson and Buonomano, 2007). An up state is a phase of persistent neuronal activity, usually a burst of action potentials lasting up to several seconds. The ensuing neuronal quiescence is termed down state. This alternation of activity and quiescence is typical of deafferented neocortex in vitro (Sanchez-Vives et al., 2010) and in vivo (Timofeev et al., 2000). A sophisticated description of the firing pattern described as cortical up and down states can be found in Holcman and Tsodyks (Holcman and Tsodyks, 2006). There the authors state, that the cerebral cortex is continuously active, even in the absence of external stimuli. An example of this spontaneous activity is the phenomenon of voltage transitions between two distinct levels, called up and down states, observed simultaneously when recording from many neurons. In the absence of sensory inputs, cortical neural networks can exhibit complex patterns of intrinsic activity. However, the origin of this spontaneous activity in the cortex is still unclear. Extracellular recordings have reported that single neuron membrane potentials in slice preparations can spontaneously transit between two states, called the up and down states. The origin of this phenomenon, originally described in (Cowan and Wilson, 1994) and (Steriade et al., 1993) is not precisely known, but the transitions have been ascribed to the intrinsic network property (Lampl et al., 1999). Transitions are almost abolished by pharmacological blockers such as glutamate receptor antagonists (Cossart et al., 2003; Shu et al., 2003a) and totally abolished by glutamate and GABAA receptor antagonists (Cossart et al., 2003). As a consequence, the up and down state transitions are a good example of how a connected network exhibits a complex correlated dynamic in the absence of any external inputs (Holcman and Tsodyks, 2006). In general, the way neuroactive drugs like general anaesthetics alter the firing patterns of cortical slice cultures – e.g. via a depression of average firing rates or a shortening of up states – yields information about their main mechanism of action (Drexler et al., 2010). But then again, if the authors would like to call that form of activity “seizure like”, I am perfectly happy with that.

Minor comments:

- What is the advantage of using the root mean voltage as a parameter to describe neuronal activity? Why is that superior to e.g. the action potential frequency or burst frequency?

- page 1, line 42: If a patient would show an increase in blood lactate during general anaesthesia I would think of a major problem, not of normal anaesthesia.

- In figure 1 there is first a slow increase in tissue oxygen, then a steep decrease followed by a second slow increase. Is this three-stage course observed in all (most of) the slices, in other words is this a pattern?

- One could consider adding a figure illustrating the effects of the combined application of rotenone and anaesthetics.

References:

Alkire MT, Hudetz AG, Tononi G (2008) Consciousness and anesthesia. Science 322:876-880.

Baars JH, Dangel C, Herold KF, Hadzidiakos DA, Rehberg B (2006) Suppression of the human spinal H-reflex by propofol: a quantitative analysis. Acta Anaesthesiol Scand 50:193-200.

Benkwitz C, Liao M, Laster MJ, Sonner JM, Eger EI, Pearce RA (2007) Determination of the EC50 amnesic concentration of etomidate and its diffusion profile in brain tissue: implications for in vitro studies. Anesthesiology 106:114-123.

Cossart R, Aronov D, Yuste R (2003) Attractor dynamics of network UP states in the neocortex. Nature 423:283-288.

Cowan RL, Wilson CJ (1994) Spontaneous firing patterns and axonal projections of single corticostriatal neurons in the rat medial agranular cortex. J Neurophysiol 71:17-32.

Crain SM, Bornstein MB (1964) Bioelectric Activity of Neonatal Mouse Cerebral Cortex during Growth and Differentiation in Tissue Culture. Exp Neurol 10:425-450.

Drexler B, Jurd R, Rudolph U, Antkowiak B (2009) Distinct actions of etomidate and propofol at beta3-containing gamma-aminobutyric acid type A receptors. Neuropharmacology 57:446-455.

Drexler B, Hentschke H, Antkowiak B, Grasshoff C (2010) Organotypic cultures as tools for testing neuroactive drugs - link between in-vitro and in-vivo experiments. Curr Med Chem 17:4538-4550.

Franks NP (2008) General anaesthesia: from molecular targets to neuronal pathways of sleep and arousal. Nat Rev Neurosci 9:370-386.

Freestone PS, Chung KK, Guatteo E, Mercuri NB, Nicholson LF, Lipski J (2009) Acute action of rotenone on nigral dopaminergic neurons--involvement of reactive oxygen species and disruption of Ca2+ homeostasis. Eur J Neurosci 30:1849-1859.

Grasshoff C, Drexler B, Hentschke H, Thiermann H, Antkowiak B (2007) Cholinergic modulation of sevoflurane potency in cortical and spinal networks in vitro. Anesthesiology 106:1147-1155.

Gredell JA, Turnquist PA, MacIver MB, Pearce RA (2004) Determination of diffusion and partition coefficients of propofol in rat brain tissue: implications for studies of drug action in vitro. Br J Anaesth 93:810-817.

Holcman D, Tsodyks M (2006) The emergence of Up and Down states in cortical networks. PLoS Comput Biol 2:e23.

Johnson HA, Buonomano DV (2007) Development and plasticity of spontaneous activity and Up states in cortical organotypic slices. J Neurosci 27:5915-5925.

Klostermann O, Wahle P (1999) Patterns of spontaneous activity and morphology of interneuron types in organotypic cortex and thalamus-cortex cultures. Neuroscience 92:1243-1259.

Lampl I, Reichova I, Ferster D (1999) Synchronous membrane potential fluctuations in neurons of the cat visual cortex. Neuron 22:361-374.

Lau PM, Bi GQ (2005) Synaptic mechanisms of persistent reverberatory activity in neuronal networks. Proc Natl Acad Sci U S A 102:10333-10338.

McCormick DA, Shu Y, Hasenstaub A, Sanchez-Vives M, Badoual M, Bal T (2003) Persistent cortical activity: mechanisms of generation and effects on neuronal excitability. Cereb Cortex 13:1219-1231.

Plenz D, Aertsen A (1996) Neural dynamics in cortex-striatum co-cultures--II. Spatiotemporal characteristics of neuronal activity. Neuroscience 70:893-924.

Rupprecht R, Papadopoulos V, Rammes G, Baghai TC, Fan J, Akula N, Groyer G, Adams D, Schumacher M (2010) Translocator protein (18 kDa) (TSPO) as a therapeutic target for neurological and psychiatric disorders. Nat Rev Drug Discov 9:971-988.

Sanchez-Vives MV, Mattia M, Compte A, Perez-Zabalza M, Winograd M, Descalzo VF, Reig R (2010) Inhibitory modulation of cortical up states. J Neurophysiol 104:1314-1324.

Shu Y, Hasenstaub A, McCormick DA (2003a) Turning on and off recurrent balanced cortical activity. Nature 423:288-293.

Shu Y, Hasenstaub A, Badoual M, Bal T, McCormick DA (2003b) Barrages of synaptic activity control the gain and sensitivity of cortical neurons. J Neurosci 23:10388-10401.

Steriade M, Nunez A, Amzica F (1993) Intracellular analysis of relations between the slow (< 1 Hz) neocortical oscillation and other sleep rhythms of the electroencephalogram. J Neurosci 13:3266-3283.

Timofeev I, Grenier F, Bazhenov M, Sejnowski TJ, Steriade M (2000) Origin of slow cortical oscillations in deafferented cortical slabs. Cereb Cortex 10:1185-1199.

Voss LJ, Brock M, Carlsson C, Steyn-Ross A, Steyn-Ross M, Sleigh JW (2012) Investigating paradoxical hysteresis effects in the mouse neocortical slice model. Eur J Pharmacol 675:26-31.

Reviewer 3 Report

This manuscript by Voss and Sleigh explores primary metabolic mechanisms, in particular inhibition of mitochondrial complex I, in the suppression of cortical synaptic function in mouse brain slices by the general anesthetics sevoflurane, ketamine and propofol. They show that the mitochondrial complex I inhibitor, rotenone, causes an increase in tissue oxygen before measurable changes in cortical population seizure-like events (SLE). Thereafter, tissue oxygen continued to be increased, accompanied by a significant and prolonged reduction of SLE RMS activity. This temporal sequence is recapitulated by the three anaesthetic drugs sevoflurane, propofol and ketamine. The authors conclude that the temporal sequence suggests that the observed “synaptic depression - as seen in anaesthesia – “is secondary to reduction in cellular metabolic capacity.

First of all, it is highly commendable that the authors are exploring a hypothesis that, as they state, “received much less attention”, and it is truly exciting to see that three general anesthetics which – in terms of neurotransmitter receptor targets – act via different receptor types have pretty similar actions in the in vitro system that the authors are using. It is less clear how what the authors are describing in train slices is related to clinical anaesthesia and whether the characterization by the authors “synaptic depression – as seen in anaesthesia” is correct.

The authors cite reference 7 as a source for the statement that most volatile and intravenous anaesthetics depress mitochondrial function. Reference 7 actually states: “In some cases (propofol, etomidate, barbiturates), it is clear that the anesthesia-inducing effects of the anesthetic [enhancement of the gamma amino-butyric acid (GABAA) receptor] are likely separate from any effect on the mitochondria (38,39). The same can be said for ketamine although the active target is different than in the former drugs (NMDA receptor).” This reviewer tends to agree with reference 7.

In the view of this reviewer, the conclusions that the author make are not fully backed by the data they are presenting, but rather speculative. The fact that the anesthetic drugs have effects similar to the mitochondrial complex I inhibitor rotenone by no means proves that they have a similar mechanisms of action. Furthermore, the question arises whether the time course of effects that the authors describe is consistent with the observed effects playing a central role in the general anesthetic effects of drugs. The time course for the latter is very short.

It is of course interesting to hypothesize how inhibition of mitochondrial complex I modulates the way in which neurotransmitter receptors or other ligand-gated ion channels respond to general anesthetics. However, the authors should explicitly acknowledge that there is direct and causal evidence that the clinical actions of at least some general anesthetics, importantly in this context of propofol are directly mediated by neurotransmitter receptors. For example, point mutations have been introduced into the beta2 [beta2-N265S] and beta3 [beta3-N265M] subunits of the GABAA receptor, which blocked clinically relevant actions of defined general anesthetics. The mutation in beta2 blocked the sedative action of etomidate [DS Reynolds et al., J Neurosci. 2003 Sep 17;23(24):8608-17], and the mutation in beta3 blunted the hypnotic action of etomidate and propofol and prevented the immobilizing action of etomidate and propofol [R Jurd et al., FASEB J. 2003 Feb;17(2):250-2]. It is not known that these point mutations would change the activity of mitochondrial complex I, but it is established that these point mutations affect modulation of GABAA receptors by selected general anaesthetics.

Round 2

Reviewer 1 Report

Manuscript: ijms-804391 Revision 1

Title: A primary metabolic mechanism for anaesthetic suppression of cortical synaptic function in mouse brain slices

Authors: Voss LJ, Sleigh JW

General.  The authors have chosen to address those issues which could be remedied by editorial changes (#1, #2, partially #3, #5).  In my opinion, they have responded adequately to each of these four points.

The other three points (#4, #6 and #7) are more problematic.  The authors state that they cannot do ATP measurements, they feel work with a mitochondrial mutant is beyond the scope of their project and they believe performing a dose response curve will be quite difficult.  So, in effect, any response that requires experimental work, they decline to undertake.  While I disagree that these experiments are too difficult for them, I do understand that they will take time and expand the conclusions beyond those they presently consider.  That is their choice.    The work is well done, I have no issues with the quality of the described work. However, in my opinion, it means that the present study remains a pilot study.

Round 3

Reviewer 1 Report

The authors have addressed all my concerns but one.  They state they are happy with the term "pilot study" but I can not find where they use it.  I would think it needs to be in both the title and abstract as well as clearly pointed out in the Discussion.

Author Response

The authors have addressed all my concerns but one.  They state they are happy with the term "pilot study" but I can not find where they use it.  I would think it needs to be in both the title and abstract as well as clearly pointed out in the Discussion.

Done